# Insecticide resistance status of indoor and outdoor resting malaria vectors in a highland and lowland site in Western Kenya

Kevin O. Owuor[1,2], Maxwell G. Machani[1], Wolfgang R. Mukabana[2,3], Stephen O. Munga[1], Guiyun Yan[4], Eric Ochomo[1], Yaw A. Afrane[5]*

1 Centre for Global Health Research, Kenya Medical Research Institute, Kisumu, Kenya, 2 School of Biological Sciences, University of Nairobi, Nairobi, Kenya, 3 Science for Health Society, Nairobi, Kenya, 4 Program in Public Health, College of Health Sciences, University of California, Irvine, California, United States of America, 5 Department of Medical Microbiology, University of Ghana Medical School, University of Ghana, Accra, Ghana

* yaw_afrane@yahoo.com

**Data Availability Statement:** All relevant data are within the paper.

**Funding:** This study was supported by grants from the National Institute of Health (R01 A1123074,

## Abstract

### Background

Long Lasting Insecticidal Nets (LLINs) and indoor residual spraying (IRS) represent powerful tools for controlling malaria vectors in sub-Saharan Africa. The success of these interventions relies on their capability to inhibit indoor feeding and resting of malaria mosquitoes. This study sought to understand the interaction of insecticide resistance with indoor and outdoor resting behavioral responses of malaria vectors from Western Kenya.

### Methods

The status of insecticide resistance among indoor and outdoor resting anopheline mosquitoes was compared in *Anopheles* mosquitoes collected from Kisumu and Bungoma counties in Western Kenya. The level and intensity of resistance were measured using WHO-tube and CDC-bottle bioassays, respectively. The synergist piperonyl butoxide (PBO) was used to determine if metabolic activity (monooxygenase enzymes) explained the resistance observed. The mutations at the voltage-gated sodium channel (*Vgsc*) gene and *Ace 1* gene were characterized using PCR methods. Microplate assays were used to measure levels of detoxification enzymes if present.

### Results

A total of 1094 samples were discriminated within *Anopheles gambiae s.l.* and 289 within *An. funestus s.l.* In Kisian (Kisumu county), the dominant species was *Anopheles arabiensis* 75.2% (391/520) while in Kimaeti (Bungoma county) collections the dominant sibling species was *Anopheles gambiae s.s* 96.5% (554/574). The *An. funestus s.l* samples analysed were all *An. funestus s.s* from both sites. Pyrethroid resistance of *An. gambiae s.l* F1 progeny was observed in all sites. Lower mortality was observed against deltamethrin for the progeny of indoor resting mosquitoes compared to outdoor resting mosquitoes (Mortality

U19 AI129326, R01 AI050243, D43 TW001505). There was no additional external funding received for this study. The funders had no role in study design, data collection and analysis, decision to publish, or preparation of the manuscript.

**Competing interests:** The authors declare that they have no competing interests.

rate: 37% vs 51%, P = 0.044). The intensity assays showed moderate-intensity resistance to deltamethrin in the progeny of mosquitoes collected from indoors and outdoors in both study sites. In Kisian, the frequency of vgsc-L1014S and vgsc-L1014F mutation was 0.14 and 0.19 respectively in indoor resting malaria mosquitoes while those of the outdoor resting mosquitoes were 0.12 and 0.12 respectively. The *ace 1* mutation was present in higher frequency in the F1 of mosquitoes resting indoors (0.23) compared to those of mosquitoes resting outdoors (0.12). In Kimaeti, the frequencies of vgsc-L1014S and vgsc-L1014F were 0.75 and 0.05 respectively for the F1 of mosquitoes collected indoors whereas those of outdoor resting ones were 0.67 and 0.03 respectively. The *ace 1* G119S mutation was present in progeny of mosquitoes from Kimaeti resting indoors (0.05) whereas it was absent in those resting outdoors. Monooxygenase activity was elevated by 1.83 folds in Kisian and by 1.33 folds in Kimaeti for mosquitoes resting indoors than those resting outdoors respectively.

## Conclusion

The study recorded high phenotypic, metabolic and genotypic insecticide resistance in indoor resting populations of malaria vectors compared to their outdoor resting counterparts. The indication of moderate resistance intensity for the indoor resting mosquitoes is alarming as it could have an operational impact on the efficacy of the existing pyrethroid based vector control tools. The use of synergist (PBO) in LLINs may be a better alternative for widespread use in these regions recording high insecticide resistance.

## Introduction

The decline in malaria incidence and prevalence have been achieved in sub-Saharan Africa through the widespread use of anti-malarial drug therapies and scaling up of vector control interventions that primarily target malaria vectors feeding and resting indoor [1]. Despite the observed achievements in malaria reduction, many parts of sub-Saharan Africa still suffer greatly from the disease [2, 3]. The recent increases in malaria transmission in many parts of sub-Saharan Africa has been partly attributed to the shifts in the mosquito biting and resting behaviours [4–7] and increasing insecticide resistance in the mosquitoes [8–10].

Insecticide resistance in malaria mosquitoes has been linked to target-site insensitivity, elevated levels of metabolic detoxifying enzymes, and behavioural resistance mechanisms [11]. Metabolic enzyme detoxification [12] and target site insensitivity [13] are responsible for higher levels of insecticide resistance [14]. Detoxification enzyme systems that have been reported to confer resistance include three major families of enzymes; the cytochrome P450 monooxygenases, esterases, and the Glutathione S-transferases. In western Kenya, about 80% of reported resistance genotypes are Vgsc-1014S *kdr* mutation, Vgsc-1014F mutations in the major vectors *Anopheles gambiae s.l.* mainly in *An. gambiae* and *Anopheles arabiensis* [15–18]. The malaria vector *Anopheles arabiensis* has been reported with increasing levels of *kdr* mutations [19]. There are no reports of *kdr* mutation at the locus 1014 in *Anopheles funestus*, also an important vector in western Kenya and many parts of Africa despite having several reports of metabolic resistance [20–22]. The increasing levels of insecticide resistance in malaria mosquitoes is believed to be mainly caused by scaling up of insecticidal treated nets (ITNs) [23, 24] and indiscriminate use of agro-chemicals for controlling crop pests in agriculture [25–27].

Field studies in East Africa have reported increased zoophagy [23, 24, 28], feeding outdoors or early evening biting [29] and changes in resting behaviour from indoor to outdoor [28, 30, 31]. These behavioural changes might have been due to selection pressure from increased coverage of LLINs [32–35]. The scale-up of LLINs in Africa has been associated with a species shift from the highly endophilic *An. gambiae* to the more exophilic *An. arabiensis* in Kenya [3, 36, 37]. The intervention pressure may selectively eliminate the most susceptible species from a population leaving the less vulnerable species able to adapt to the new environment [38]. While the majority of studies have reported the existence of insecticide resistance and the mechanisms involved, there is a paucity of detailed information on the association of insecticide resistance in malaria vectors with the observed resting behavior in the field.

Malaria transmission is dependent on the propensity of malaria vectors to feed on human hosts and preference to live in close proximity to human dwellings [7]. Given the importance of mosquito feeding and resting behaviour to the successes of malaria vector control and transmission, it is important to understand the influence of physiological resistance on the resting behaviour of malaria vectors and how the observed behaviours could impact the effectiveness of the existing frontline interventions. Currently, the mechanisms underlying the observed behavioural shifts in malaria vectors are poorly known, and it may have an epidemiological consequence. In order to maintain the efficacy of insecticide-based vector control, insecticide resistance should be constantly monitored and management strategies developed and deployed [8, 39–43]. The present study attempts to answer how insecticide use and resistance influences resting behaviours and reports on the status of insecticide resistance and mechanisms involved in indoor and outdoor resting malaria vectors.

## Methods

### Study sites

The study was carried out in the lowland site of Kisian (00.0749˚ S, 034.6663˚ E, altitude 1,137–1,330 m above sea level) in Kisumu county and the highland site of Kimaeti (00.6029˚ N, 034.4073˚ E, altitude 1,430–1545 m above sea level) in Bungoma county all in Western Kenya. These sites have high abundance of malaria mosquitoes (*An. gambiae s.l.* and *An. funestus s.l.*) and high level of insecticide resistance [15, 17]. Kimaeti (Bungoma county) has extensive tobacco cultivation visible by large farms with numerous curing kilns observed within the village in the region. In Kisian (Kisumu county), there is sand harvesting from river beds, fishing, rice and maize farming most of which enhance mosquito breeding habitats. There is extensive use of agrochemicals on these farms which could have a potential role in the mediation of resistance to insecticides [44]. Western Kenya experiences long rainy seasons between the months of March to June and the short rainy seasons between the months of October and November [45].

### Mosquito sampling

Resting *Anopheles* mosquitoes were sampled indoors and outdoors from household units. Mosquito collections were made during the long rainy season (May-July) and the short rainy season (October-November) of 2019. Thirty (30) houses were randomly selected per site and resting mosquitoes collected from 06:00 to 09:00 h both indoor and outdoor resting points. For indoor resting mosquitoes, a Prokopack aspirator (JohnWHock, Gainesville, FL, USA) and mouth aspirator were employed to collect mosquitoes indoors. Briefly, collections were done by hovering the aspirator systematically over the walls up and down, under the furniture and on hanged clothing for about 1 minute per second [46, 47]. Outdoor collections were sampled from pit shelters dug (1.5M×1.5M×1.5M) in the ground constructed according to

Muirhead-Thomson's method [48], from clay pots or containers placed at least 10 meters outside of houses and from any proximal human outdoor resting points such as granaries, outdoor kitchen, under shaded places and evening outdoor human resting points. Sampled anophelines were first discriminated using morphological keys [49]. Further species-specific identification within the *An. gambiae s.l.* and *An. funestus s.l.* was conducted using PCR. Mosquito collections were done at the beginning and at the end of the dry and rainy seasons. The samples collected were taken to the entomology laboratories at the Kenya Medical Research Institute (KEMRI), Center for Global Health Research (CGHR) for subsequent rearing, phenotypic, biochemical and molecular analyses.

## Rearing of mosquitoes

Blood-fed and half-gravid female *Anopheles* mosquitoes from both the indoor and outdoor collections were aspirated into separate labeled netted mosquito holding cages measuring 30cm × 30cm × 30cm where they were maintained at 25 ± 2˚C and relative humidity of 80 ± 4% with 12:12 hours of light and dark. They were provided with 10% sucrose solution imbibed in cotton wool. Oviposition cups were introduced into the cages for egg collection. Since all collections made were put together in similar cages, the number of mosquitoes that laid eggs was not determined. Eggs collected were transferred into larval rearing trays containing spring water where they hatched. The aquatic larval stages were maintained in water 26–27˚C and were fed on a mixture of Tetramin™ fish food and brewer's yeast. After the four larval stages, pupae were picked and transferred into netted holding cages in small cups where the emergent adults were provided with 10% sucrose solution [50].

## Testing phenotypic resistance in the F1 progeny of indoor and outdoor resting mosquitoes

First filial generation (F1) females raised from field-collected adults that were resting either indoors or outdoors, that were 3–5 -day old, were tested for susceptibility using the standard WHO tube bioassays (WHO, 2016) against discriminating doses of four insecticides selected from two classes: (i) Pyrethroids—(0.05% deltamethrin, 0.75% permethrin and (0.05% Alphacypermethrin); and (ii) organophosphate—(5% malathion). For each test about 100–150 mosquitoes were used for the assay comprising 20–25 mosquitoes for each of four replicates for each of the insecticides and controls. Silicone oil-treated papers were used as a control for pyrethroid assays while olive oil was used for the malathion (organophosphate) test. Mosquitoes were exposed for 1hour for each insecticide and the number that were knocked down recorded after every 10 mins within the 1-hour exposure period. After 1-hour exposure to the diagnostic concentrations, mosquitoes were transferred to recovery cups and maintained on 10% sucrose solution for 24 hrs. Mosquito survival status was examined at 24-hour post-exposure, where the survived and dead mosquitoes were collected and preserved at -20˚C prior to molecular analysis. Percentage mortality was calculated for both indoor and outdoor F1 mosquitoes.

## Piperonyl butoxide (PBO) synergist bioassays

The involvement of oxidase (P450) resistance mechanism in pyrethroid resistance was determined by pre-exposing test populations to the oxidase inhibitor; Piperonyl butoxide synergist (PBO). Briefly, unfed females aged 3–5 days were pre-exposed to 4% PBO impregnated test papers for one hour. After pre-exposure to PBO, the mosquitoes were immediately exposed to each of the three pyrethroids (deltamethrin, permethrin and alphacypermethrin) separately for another hour. One batch of 25 females was only exposed to 4% PBO without insecticide as

a control. Mosquitoes were transferred to holding tubes and supplied with 10% sugar solution. Mortality was recorded after 24 hour recovery period.

## Measurement of insecticide resistance intensity in the F1 progeny

Insecticide resistance intensity testing to deltamethrin was determined by using CDC bottle bioassay with serial dosages. Serial concentrations (1×, 5× and 10×) of deltamethrin were prepared and used for the CDC bottle assays. The bottles were coated in batches for each working concentration, to which mosquitoes were exposed as per the CDC procedure guide MR4 [50, 51]. The number of knocked-down mosquitoes was recorded every 10 minutes until either all mosquitoes in the test bottles were dead or it reached 1 hour after the start of the experiment. Mosquitoes were transferred to holding cups and fed on 10% sucrose solution. Mortality was recorded after 24-hours.

## Molecular identification and genotyping of resistance alleles

Genomic DNA was extracted by the alcohol precipitation method and conventional PCR was used to speciate the samples [50, 52, 53]. The taqMan assay was used to detect the mutations (*Vgsc*-1014S, *Vgsc*-1014F and *N1575Y*) at the voltage-gated sodium channel [54, 55] and the same set of samples were used to detect the *G119S* mutation in *Ace 1* [56].

## Biochemical enzyme levels in F1 progeny of indoor and outdoor resting *An. gambiae s.l.*

From both sites, indoor and outdoor, 100-three-day old female mosquitoes, were killed by freezing for 10 minutes and homogenized individually in 0.1 M potassium Phosphate ($KPO_4$) buffer as described by Benedict, (2014). The levels of metabolic enzymes; β-esterases, Glutathione S-transferase (GST) and Oxidases were measured using microplate enzyme assays. To correct for variations in mosquito sizes, the protein content of each mosquito was measured by adding 20μl of mosquito homogenate to the microtiter plates in triplicates and 80μl of $KPO_4$ to each well after which 200μl of protein-dye reagent was toped up. A standard curve was used to relate amount of protein used. The absorbances were taken using a microplate reader [50, 57, 58].

## Ethical considerations

Ethical approval for the study was obtained from Ethical Review Board of Kenya Medical Research Institute under number SERU 3613. Permission was sought from community leaders of each study site. Informed consent was obtained from the household heads. For mosquito larvae collection, oral consent was obtained from field owners in each location. These locations were not protected land, and the field studies did not involve endangered or protected species.

## Data analysis

The phenotypic resistance assays were expressed as proportions of mortality around 95% confidence interval and classified by WHO (2016) as a guide. Genotypic data for species identification was weighted as proportions of the samples assessed. The allele frequencies for resistant genotypes were calculated using the Hardy-Weinberg equilibrium equation. Metabolic resistance enzymes were analyzed by ANOVA after which the source of variation between the fold changes was determined by the Turkey-Kramer HSD test. All statistical analyses were done in R software version 3.6.3.

## Results

### Species discrimination of *An. gambiae s.l.* and *An. funestus s.l.*

A total of 1074 samples were identified to species within the *An. gambiae s.l.* and 289 from the *An. funestus s.l.* from the two sites. In the lowland site of Kisian (Kisumu county), out of 500 *An. gambiae s.l.* samples analysed, *An. arabiensis* composition was 74.2% (95% CI; 71.5–78.9%) while *An. gambiae s.s.* was 25.8% (95% CI; 21.1–28.5%). All 122 *An. funestus s.l* samples analysed from indoors were *An. funestus s.s.* (Table 1). In the highland site of Kimaeti (Bungoma county) out of 574 *An. gambiae s.s.* composition was 96.5% (95% CI; 95.0–98.0%) while *An. arabiensis* was 3.5% (95% CI; 2.0–5.0%). The 167 *An. funestus s.l.* analysed were all *An. funestus s.s.* (Table 1).

### Phenotypic resistance in the F1 progeny of indoor and outdoor mosquitoes

A total of 2,800 female *An. gambiae s.l.* (Kisian = 1,400 and Kimaeti = 1,400) and 1,600 female *An. funestus s.l.* (Kisian = 800 and Kimaeti = 800) were used in the WHO tube assays. In the lowland site of Kisian, the mortality rate of the indoor resting *An. gambiae s.l.* mosquitoes exposed to deltamethrin was significantly lower than outdoors resting ones (37%, 95% [CI; 28–46%]) vs 51% [95% CI; 41–61%] respectively; t = 2.035, df = 6, P = 0.044). The indoor resting *An. gambiae s.l.* had significantly lower mortality rate to permethrin than those resting outdoors (31% [95% CI; 22–40%] vs 51% [95% CI; 41–61%], t = 2.078, df = 6, P = 0.042). Following exposure to alphacypermethrin, the mortality rate for indoor resting *An. gambiae s.l.* was 30% (95% CI; 21–39%) compared to their outdoor counterparts with 60% (95% CI; 50–70%) (t = 4.392, df = 6, P<0.05). There was 100% mortality for both the indoor resting and outdoor resting *Anopheles gambiae s.l.* when exposed to malathion (Fig 1A).

Indoor resting F1 progeny raised from *Anopheles gambiae s.l.* collected from the highland site of Kimaeti had a mortality rate of 49% (95% CI; 39–59%) compared to those resting outdoors 53% (95% CI; 43–63%) when exposed to deltamethrin. Although the indoor resting mosquitoes showed a slightly lower mortality rate compared to outdoors, this was not statistically significant (t = 0.474, df = 6, P>0.05). Exposure of mosquitoes to permethrin showed for indoor resting mosquitoes had a significantly lower mortality 7% (95% CI; 1–12%) compared to those resting outdoors 51% (95% CI; 41–61%), (t = 6.063, df = 6, P<0.001). Mosquitoes exposed to alphacypermethrin on the other hand showed a mortality rate of 70% (95% CI; 61–79%) for indoor resting mosquitoes compared to those resting outdoors outdoors80% (95% CI; 72–88%), though this was not significantly different (t = 1.058, df = 6, P>0.05). Exposure of mosquitoes from the indoor or outdoor location in showed that *An gambiae s.l.* were fully susceptible to malathion with 100% mortality (Fig 1A).

**Table 1. *An. gambiae s.l.* and *An. funestus s.l.* species composition from indoor and outdoor resting collections from Western Kenya.**

| Site | | Anopheles gambiae s.l | | | Anopheles funestus s.l. |
|---|---|---|---|---|---|
| | Location | An. gambiae s.s (%) | An. arabiensis(%) | Total | An. funestus s.s |
| **Kisian** | Indoor | 83(33.2) | 167(66.8) | 250 | 122(100) |
| | Outdoor | 46(18.4) | 204(81.6) | 250 | 0 |
| | **Total** | **129(25.8)** | **371(74.2)** | **500** | **122(100)** |
| **Kimaeti** | Indoor | 304(99.1) | 3(0.9) | 307 | 167(100) |
| | Outdoor | 250(93.6) | 17(6.4) | 267 | 0 |
| | **Total** | **554(96.5)** | **20(3.5)** | **574** | **167(100)** |

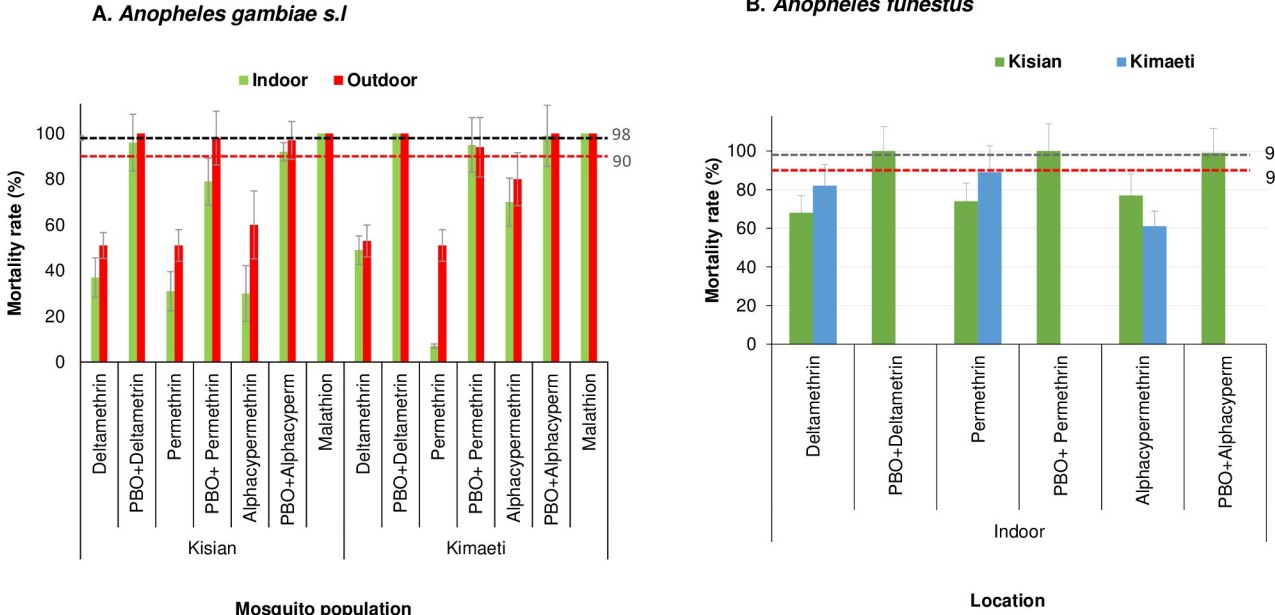

**A. *Anopheles gambiae s.l***

**B. *Anopheles funestus***

**Fig 1. Percentage mortality rates for indoor and outdoor resting A.) *An gambiae s.l* B.) *An funestus* F1 progeny from Kisian (lowland) and Kimaeti (Highland) using WHO tube bioassays.** Error bars indicate 95% confidence intervals. The 90% mortality threshold for declaring suspected resistance and 98% mortality threshold for calling full susceptibility based on the WHO criteria are indicated.

Addition of PBO synergist to the test, partially restored the resistance of indoor resting mosquitoes from 37% to 96% for deltamethrin (t = 9.0, df = 6, P<0.001), 31% to 79% permethrin (t = 5.908, df = 6 P = 0.005) and 30% to 92% for alphacypermethrin (t = 8.598, df = 6, P<0.001) in Kisian. The effects of the PBO synergist was evident in the outdoor resting mosquitoes with mortality rate range; 98%-100% for the three pyrethroids used, confirming the full involvement of monooxygenase enzyme activity in the pyrethroid detoxification (Fig 1A).

In Kimaeti, the addition of PBO to tests involving indoor resting *An. gambiae s.l.* showed significantly increased mortality rate from 49% to 100% (t = 7.095, df = 6, P<0.001) for deltamethrin, 7% to 95% (t = 16.436, df = 6, P<0.001) for permethrin and 70% to 99% (t = 5.385, df = 6, P = 0.001) for alphacypermethrin. The effects of the PBO synergist was also seen in outdoor resting mosquitoes with the mortality rate ranging between 94% and 100% (Fig 1A).

Due to the small number collected outdoors and the general difficulty in raising the F1, only indoor *An. funestus s.l.* from both study sites were assayed. In Kisian, the mortality rate of *An. funestus* was 68% (95% CI; 59–77%) to deltamethrin, 74% (95% CI; 65–83%) to permethrin and 77% (95% CI; 69–85%) to alphacypermethrin (Fig 1B). In Kimaeti, the F1 of *An. funestus* showed mortality rates of 62% (95% CI; 52–72%) when exposed to deltamethrin, 89% (95% CI; 83–95%) to permethrin and 61% (95% CI; 51–71%) following alphacypermethrin exposure. There was 100% mortality across both sites with PBO pre-exposure (Fig 1B).

## Intensity of insecticide resistance in F1 of *An. gambiae s.l.* resting indoors and outdoors

The mortality rate for indoor *An. gambiae s.l.* from Kisian that were exposed to 1×, 5× and 10× of the diagnostic doses of deltamethrin was 42% (95% CI; 32–52%), 78% and 100% respectively whilst for outdoors was 51% (95% CI; 41–61%), 83% (95% CI; 76–90%) and 100%, indicating moderate-intensity resistance across both locations according to the WHO 2016 criteria [59]

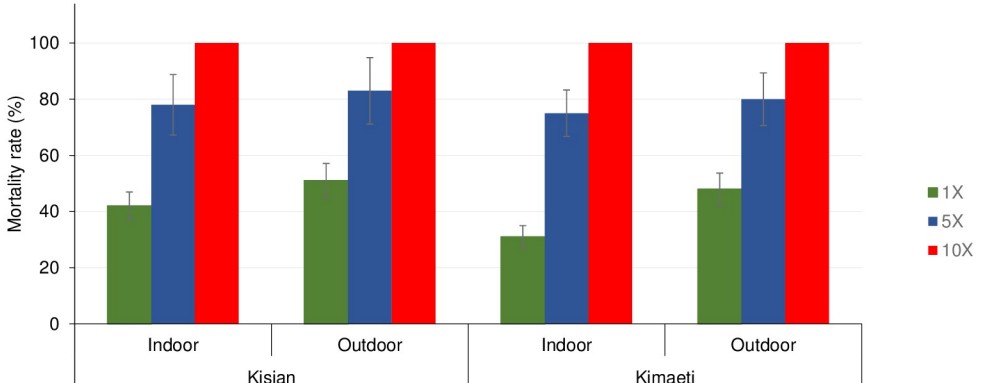

**Fig 2. Mortality rates of *An. gambiae s.l* F1 progeny from indoor and outdoor resting collections recorded using CDC bottle intensity assays.** Error bars indicate 95% confidence intervals. The 90% mortality threshold for declaring suspected resistance and 98% mortality threshold for calling full susceptibility based on the WHO criteria are indicated.

(Fig 2). Although there was lower mortality among the indoor resting mosquitoes compared to their outdoor counterparts at 1× (t = 1.269, df = 6, P = 0.130) and at 5× (t = 0.823, df = 6, P = 0.221), this was not statistically significant (Fig 2).

The mortality rate of indoor resting population from Kimaeti exposed to 1×, 5× and 10× concentration of deltamethrin were 31% (95% CI; 22–40%), 75% (95% CI; 67–83%) and 100% respectively while the outdoors were 48% (95% CI; 38–58%), 80% (95% CI; 72–88%) and 100% respectively indicating moderate-intensity resistance in both locations according to the WHO 2016 criteria [59]. Similarly, even though the mortality rates were lower indoors than outdoors, there was no significant statistical difference between the two populations at 1× (t = 1.512, df = 6, P>0.05) and at 5× (t = 0.808, df = 6, P>0.05) (Fig 2).

## Target site genotyping for resistance alleles in the F1 of indoor and outdoor resting *An. gambiae s.l.*

In Kisian, the frequency of the vgsc L1014S and L1014F in the progeny of mosquitoes resting indoors were present with frequencies of 0.14 and 0.19 respectively for the F1of indoor resting mosquitoes whereas those raised from mosquitoes resting outdoors were 0.14 and 0.12 respectively. The *ace 1* mutation was present by higher frequency in the F1 of mosquitoes resting indoors (0.23) compared to those of the ones resting outdoors (0.12). The vgsc-1014S and *ace 1* mutations were not observed in *An. gambiae* from Kisian due to the small sample size.

The frequency of L1014S and L1014F present in mosquitoes collected indoors were 0.75 and 0.05 respectively in Kimaeti compared to those raised from mosquitoes collected outdoors (0.67 and 0.03 respectively). The ace 1 G119S mutation was observed in the F1 of mosquitoes resting indoors with a frequency of 0.05 and was not present in those of mosquitoes resting outdoors. The *kdr* point mutation at locus 1575Y was not present in both study sites (Table 2).

## Biochemical enzyme levels in F1 progeny of indoor and outdoor resting *An. gambiae s.l.*

The monooxygenases, β-Esterase and Glutathione S-transferases activities were analyzed to determine the level of involvement in the F1 of *An. gambiae s.l.* insecticide resistance. In Kisian, the monooxygenase activity was increased by 1.83 folds in the progeny of *An. gambiae*

**Table 2. Frequency of resistant alleles (*Kdr* and *Ace1-G119S*) in indoor and outdoor-resting *An. gambiae s.s* and *An. arabiensis* populations from Western Kenya.**

| Site | Location | Species | n | Vgsc (*kdr*) | | | Ace 1 |
|------|----------|---------|---|--------------|---|---|-------|
| | | | | Locus 1014 | | Locus 1575 | Locus 119 |
| | | | | L1014S | L1014F | 1575Y | G119S |
| Kisian | Indoor | *An.gambiae* | 8 | 0 | 0.25 | 0 | 0 |
| | | *An.arabiensis* | 36 | 0.14 | 0.19 | 0 | 0.23 |
| | Outdoor | *An.gambiae* | 1 | 0 | 0 | 0 | 0 |
| | | *An.arabiensis* | 43 | 0.14 | 0.12 | 0 | 0.12 |
| | Total | *An. gambiae* | 9 | 0 | 0.33 | 0 | 0 |
| | | *An.arabiensis* | 79 | 0.08 | 0.06 | 0 | 0.19 |
| Kimaeti | Indoor | *An.gambiae* | 43 | 0.75 | 0.05 | 0 | 0.05 |
| | | *An.arabiensis* | 1 | 0.01 | 0 | 0 | 0 |
| | Outdoor | *An.gambiae* | 39 | 0.67 | 0.03 | 0 | 0 |
| | | *An.arabiensis* | 5 | 0.60 | 0 | 0 | 0 |
| | Total | *An.gambiae* | 82 | 0.72 | 0.06 | 0 | 0.02 |
| | | *An.arabiensis* | 6 | 0.07 | 0 | 0 | 0 |

*s.l.* resting indoors and by 1.66-folds for those resting outdoors when compared to the insectary reference Kisumu strain ($F_{2,134}$ = 105.20, P<0.05, Fig 3A). The β-Esterases fold change was not significantly different between F1 progeny raised from indoor and outdoor resting *An. gambiae s.l.* mosquitoes ($F_{2,134}$ = 188.50, P<0.05, Fig 3B). In Kisian, the elevation of GSTs was by a 2.3-fold change in the F1 of indoor-resting mosquitoes which was significantly higher than that of the F1 of those resting outdoors ($F_{2,134}$ = 95.14, P<0.05, Fig 3C).

The enzyme activity of monooxygenases was higher by 1.3-fold in the indoor population from Kimaeti compared to the outdoor population ($F_{2,134}$ = 51.43, P<0.05, Fig 3A). The activity of β-esterases from Kimaeti was elevated by 1.2 folds for the indoor-resting population which was significantly different compared to that of the outdoor resting mosquitoes ($F_{2,134}$ = 36.66, P<0.001, Fig 3B). The activity of Glutathione S-transferase was elevated by a 3.0-fold change in the progeny of mosquitoes found resting indoors than those found resting outdoors ($F_{2,134}$ = 119.9, P<0.05) (Fig 3C).

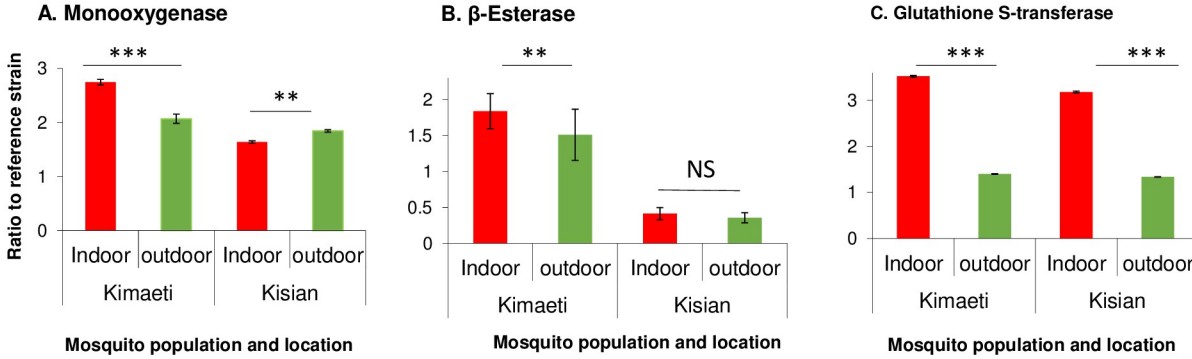

**Fig 3. Metabolic enzyme activity for indoor and outdoor resting F1 progeny of *An. gambiae* from Kisian and Kimaeti in Western Kenya.** A: monooxygenases; B: β-esterases; and C: Glutathione S-transferase. Enzyme activities were expressed as the ratio of a population of interest to the Kisumu reference strain. Error bars indicates 95% confidence intervals. *, P < 0.05; ***, *P* < 0.001; NS; not significant.

## Discussion

This study set out to determine the level of insecticide resistance of *Anopheles* mosquito species between populations found resting indoors and those resting outdoors. Generally, high phenotypic, physiological (genotypic and metabolic) resistance was observed in the progeny of indoor resting malaria mosquitoes than the outdoor resting vectors.

In the lowland sites of Kisian (Kisumu county), *An. arabiensis* was the most abundant malaria vector compared to its sibling species *An. gambiae s.s.* whereas in Kimaeti (Bungoma county), the dominant species was *An. gambiae s.s.* similar to earlier reports [17, 21, 28, 60, 61]. The lowlands tend to have high temperatures and low humidity which favour the more resilient *An arabiensis* whereas in the highlands, there are low temperatures and high relative humidity which favour *An gambiae* [62].

The indoor population recorded high phenotypic resistance to pyrethroids than outdoors. The phenotypic insecticide resistance to pyrethroids in *An. gambiae s.l.* is widespread in Western Kenya evident in previous studies [15, 17, 19]. The resistance to pyrethroids by *An. funestus* was observed and has as well been reported before [63]. These regions of Western Kenya have been reported to have increasing resistance to pyrethroids which are the public health approved insecticides for use in LLINs [15, 17, 20, 42]. There was 100% susceptibility to malathion of mosquitoes just as similar studies have shown in Ghana [64]. Synergist PBO pre-exposure restored susceptibility for both indoor and outdoor resting mosquitoes, revealing the role of detoxifying metabolic enzymes in the insecticide resistance in these regions. This means, therefore, that there are more factors at play contributing to the insecticide resistance present in Western Kenya similar to studies before [12, 65, 66]. Increasing the concentration of the deltamethrin in CDC bottle assays restored susceptibility to 100% suggesting that the continuous exposure to the current dosage in LLINs and possible interaction with non-lethal doses in agricultural chemicals could have been at play to contribute to the development of resistance to pyrethroids as previously demonstrated [38] in indoor resting and outdoor resting malaria mosquitoes. The result showed moderate intensity insecticide resistance since the mosquitoes succumbed to the highest concentration according to the WHO test procedures for insecticide resistance monitoring in malaria vectors [59]. The buildup of the phenotypic resistance which was higher in indoor resting mosquitoes compared to the outdoor resting counterparts might be threatening current insecticide-based malaria control interventions as suggested by prior studies [67, 68].

The presence of resistance-associated point mutations was more in indoor resting mosquitoes than their outdoor resting counterparts. This can be attributed to the adaptations from selection pressures due to constant exposure to insecticide-based interventions such as LLINs [17, 23, 39, 69] and the extensive chemicals used in the tobacco farms in Kimaeti. The study also detected, even though in lower frequencies, a significant proportion of the vgsc-1014S and 1014F in *An. arabiensis* a phenomenon that has been previously reported [17, 19, 66]. This is in line with studies that have shown the occurrence of more than one *kdr* associated point mutation within a population of *An. gambiae s.l.* already reported previously [17, 20, 61, 66, 70]. The significant vgsc mutations observed could be a result of selection pressure build-up that is due to more contact with insecticides in indoor-based interventions [17, 39, 42, 61, 66]. From Kisian, the *G119S* mutation was present at low frequencies even though it was higher in the progeny of mosquitoes resting indoors compared to those resting outdoors. This was more in Kisian, where the vgsc mutations were at lower frequencies than in Kimaeti. These findings suggest that these mutations could be arising from different pressures that could be present in the lowland and absent in the highland.

The metabolic enzymes, associated with insecticide resistance (monooxygenases, β-esterases, and glutathione S-transferases) activities were found to be elevated, more in indoor resting malaria mosquitoes compared to the outdoor counterparts from both sites. From the phenotypic assays, pre-exposure to PBO synergist restored the susceptibility of the malaria vectors to the pyrethroids commonly used in LLINs by public health. Phenotypic exposures with prior PBO contact demonstrated more activity of monooxygenases in aiding metabolic resistance. The involvement of monooxygenases in pyrethroid resistance has been reported in Western Kenya [17]. In Kimaeti, there was increased levels β-esterases, higher indoors than outdoors. Kisian, on the other hand, did not show involvement of β-esterases in contributing to resistance as shown by similar levels in indoor and outdoor resting mosquitoes. The glutathione-S-transferase possibly played a part in the resistance levels as a previous study reported [71] since it was higher in mosquitoes resting indoors than those resting outdoors from both Kisian and Kimaeti. These levels, therefore, suggest that monooxygenases were the main mechanism of insecticide resistance in Kisian, especially with the low frequency of resistant alleles, whereas in Kimaeti, the case pointed be a combination of genotypic and metabolic mechanisms.

The expression of phenotypic, genotypic and metabolic resistance appears to be higher in indoor than outdoor resting malaria mosquitoes in these regions. The widespread use of LLINs in attempts to controlling these vectors and the extensive agrochemical use could be strengthening the increase of insecticide resistance in the sites [21, 61]. The higher levels indoors suggest that these mosquitoes could be resting indoors because they are adequately resistant to the insecticides used in LLINs, posing a threat to the wide coverage LLINs [21]. On the other hand, outdoors, the resistance mechanisms were present as well pointing to exposure to these insecticide-based interventions in just enough pressure to elicit expression of the resistance traits. The levels of resistance could be enough to elicit an increase in malaria incidence due to the reduced mortality of resistant malaria vectors that could hinder current vector control interventions [67].

## Conclusion

In this study there was high phenotypic, genotypic and metabolic insecticide resistance in indoor resting malaria vectors (*An. gambiae s.l* and *An. funestus*) compared to outdoor-resting mosquitoes. Indoor-based insecticide control interventions are potentially at the verge of becoming obsolete due to the reduced efficacy in controlling resistant malaria vectors which in turn might lead to rise in malaria incidence. This calls for urgent improvement of these interventions and development of alternative tools for indoor malaria control coupled with strengthening of insecticide resistance monitoring. The use of synergist (PBO) in LLINs may be a better alternative for widespread use in these regions recording high insecticide resistance.

## Acknowledgments

The authors wish to thank the villagers and community leaders for their permission to collect mosquitoes in their houses. In particular, we thank the community leaders. We acknowledge the Entomology Laboratory at Kenya Medical Research Institute, Kisumu, more specific Joyce K. Osoro, Mathew Kipsum, Christabel A. Winter, Duncan O. Tindi and Benard O. Alele for providing technical support and laboratory space for the study. This paper is published with permission from the director of Kenya Medical Research Institute.

## Author Contributions

**Conceptualization:** Guiyun Yan, Eric Ochomo, Yaw A. Afrane.

**Data curation:** Kevin O. Owuor, Maxwell G. Machani.

**Formal analysis:** Kevin O. Owuor, Maxwell G. Machani.

**Funding acquisition:** Guiyun Yan, Yaw A. Afrane.

**Supervision:** Wolfgang R. Mukabana, Stephen O. Munga, Guiyun Yan, Eric Ochomo, Yaw A. Afrane.

**Validation:** Wolfgang R. Mukabana, Guiyun Yan, Eric Ochomo, Yaw A. Afrane.

**Writing – original draft:** Kevin O. Owuor, Maxwell G. Machani.

**Writing – review & editing:** Maxwell G. Machani, Wolfgang R. Mukabana, Stephen O. Munga, Guiyun Yan, Eric Ochomo, Yaw A. Afrane.

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
