## [Decision Letter · Decision Letter 0]

10 Dec 2020

PONE-D-20-30800

Insecticide resistance status of indoor and outdoor resting malaria vectors in a highland and lowland site in Western Kenya

PLOS ONE

Dear Dr. Afrane,

Thank you for submitting your manuscript to PLOS ONE. After careful consideration, we feel that it has merit but does not fully meet PLOS ONE’s publication criteria as it currently stands. Therefore, we invite you to submit a revised version of the manuscript that addresses the points raised during the review process.

Please address the comments raised by the reviewers, and pay careful attention to the issue raised by reviewer 2 concerning the validity of the data with respect to the way the mosquitoes were collected i.e. is there clear differentiation between indoor and outdoor resting mosquitoes based on the way they were collected? 

Please submit your revised manuscript within the next 2 months. If you will need more time than this to complete your revisions, please reply to this message or contact the journal office at plosone@plos.org. Please include the following items when submitting your revised manuscript:

We look forward to receiving your revised manuscript.

Kind regards,

Basil Brooke, PhD

Academic Editor

PLOS ONE

Journal Requirements:

"This study was supported by grants from the National Institute of Health

Y.A Grant number R01 A1123074, U19 AI129326,

G.Y R01 AI050243, D43 TW001505

The funders had no role in the study design, data collection and analysis, decision to publish or preparation of the manuscript.".

i) Please provide an amended statement that declares *all* the funding or sources of support (whether external or internal to your organization) received during this study, as detailed online in our guide for authors at http://journals.plos.org/plosone/s/submit-now.  Please also include the statement “There was no additional external funding received for this study.” in your updated Funding Statement.

ii) Please include your amended Funding Statement within your cover letter. We will change the online submission form on your behalf.

3.  We note that Figure  [1] in your submission contains a map image which may be copyrighted. All PLOS content is published under the Creative Commons Attribution License (CC BY 4.0), which means that the manuscript, images, and Supporting Information files will be freely available online, and any third party is permitted to access, download, copy, distribute, and use these materials in any way, even commercially, with proper attribution. For these reasons, we cannot publish previously copyrighted maps or satellite images created using proprietary data, such as Google software (Google Maps, Street View, and Earth). For more information, see our copyright guidelines: http://journals.plos.org/plosone/s/licenses-and-copyright.

1.     You may seek permission from the original copyright holder of Figure [1] to publish the content specifically under the CC BY 4.0 license.  

Reviewers' comments:

Reviewer's Responses to Questions

**Comments to the Author**

1. Is the manuscript technically sound, and do the data support the conclusions?

Reviewer #1: Yes

Reviewer #2: Partly

2. Has the statistical analysis been performed appropriately and rigorously? 

Reviewer #1: Yes

Reviewer #2: No

3. Have the authors made all data underlying the findings in their manuscript fully available?

Reviewer #1: Yes

Reviewer #2: Yes

4. Is the manuscript presented in an intelligible fashion and written in standard English?

Reviewer #1: No

Reviewer #2: No

5. Review Comments to the Author

Reviewer #1: My comments are below

Introduction

Line 73 to 75 « Despite the observed improvement in malaria incidence and prevalence in many parts of sub-Saharan Africa, transmission is increasing in several countries [2,3]. » I don’t think the term « improvement » is appropriate and the sentence doesn’t make sense

Line 80 to 81 « Insecticide resistance in malaria mosquitoes is linked to presence and increase in metabolic detoxification enzymes, target site insensitivity and behavioural resistance [11]. » the sentence is not clear can the author rephrase the sentence to improve understanding?

Line 84 to 86 « Detoxification enzyme systems that have been reported to confer resistance include three major families of enzymes; the cytochrome P450 monooxygenases, β-esterases, and the Glutathione S-transferases. » please write esterase as you are refering to enzymes families

Line 92 to 95 « The increasing levels of insecticide resistance in malaria mosquitoes have been associated with continuous exposure to insecticides in Long Lasting Insecticide Nets( LLINs) [23,24] and agro-chemicals such as pesticides due to the creation of selection pressures [25-27]. » The phrase is not clear and need to be rewritten to improve understanding

The introduction is not straight forward and is poorly written the authors need to revise this section carefully to improve understanding

Methods

Study sites

Line 120 « The study was carried out in the lowland site of Kisian (0.0749° S, 34.6663° E, 1,137m) in Kisumu county … » The geographical coordinates is not appropriate can the authors check for the correct coordinates ?

Line 127 to 128 « There is extensive use of agrochemicals on these farms which could have a potential role in the mediation of resistance to insecticides. » Are anopheline breeding habitats found in tobacco farms and what type of insecticides are used in these farms please provide more information. Please also say how frequent insecticides are sprayed in these farms. How distant are these farms from the village.

Mosquito sampling

Mosquitoes were collected in how many houses per site and outdoor places please include information

Line 156 to 160 « First filial generation (F1) females raised from field-collected adults that were resting either indoors or outdoors, that were 3-5 -day old, were tested for susceptibility using the standard WHO tube bioassays (WHO, 2016) against discriminating doses of five insecticides selected from three classes: (i) Pyrethroids - (0.05% deltamethrin, 0.75% permethrin and (0.05% Alphacypermethrin); and (ii) organophosphate - (5% malathion). » The author say they tested 5 insecticides only a list of four insecticide is provided please correct.

Line 166 to 167 « Mortality was defined as the inability of the mosquitoes to stand or to fly in a coordinated manner. » is this mortality or knock down effect please correct

Line 175 to 176 « After pre-exposure to PBO, the mosquitoes were immediately exposed to the three pyrethroids (deltamethrin, permethrin and alphacypermethrin) for an additional hour. » Where mosquitoes exposed to all three insecticides at once or seperately to each insecticide please clarify

Results

There are data missen in this part the author need to give the number of gravid and non gravid females collected outdoor and indoor in each site. They could also indicate the number collected during each season.

The number of mosquitoes tested for each insecticide need to be provided

Table 1

The authors say that the percentage is in bracket but it is not clear how they did the calculations of the percentage of An. funestus in Kisian 122(19%) in Kimaeti they also wrote 167 (23.16) can they provide explanation for their calculations or correct.

The author also need to check their calculations the row on An. arabiensis in Kisian the total in the column and line also need to be corrected.

Reviewer #2: Comments:

This study intends to show the insecticides resistance between indoors and outdoors resting vectors in highlands and lowlands

This study has a major drawback in methodology.

To start with the ecological behavior of An. gambiae s.l is to feed indoor and rest indoor for An. gambiae s.s. while for An. arabienses is feeding indoor and resting outdoor or feed outdoor and rest outdoors. Due to the wide coverage of the insecticide treated surfaces (LLINs and RS), mosquitoes have changed the feeding and resting behavior to forfeit the effect of the insecticides treated surfaces. Due to this, higher proportions have found to feed and rest outdoors regardless of the vector species.

Sampling:

The samplings of vectors were done indoor and outdoor. The vector after feeding indoor they get out to rest and search for oviposition site. This is indeed a challenge to mark that this mosquito was found resting outdoor while it fed indoor. The physiological and enzymatic dynamics in mosquitoes is mostly reflected by blood meal source. There is no barrier between indoor and outdoor resting populations hence free mating is possible. In first place, authors should be aware that resistance is genetical and is clearly related age of the mosquito. These mosquitoes share breeding sited and feeding sources. The emerging young mosquitoes resting outdoor before searching for host. The outdoor collection is probably that it collected emerged and gravid mosquitoes searching for oviposition site. The methods were weak not to show the separation of population fed outdoor or indoor. So collection alone cannot provide a guarantee of the biochemical processes to be elevated by resting position.

I personally reject this paper as the basis used of outdoor and indoor resting have no barrier separation between the populations collected. Since resistance is genetical and biochemical process is influenced by blood meal sources. No evidence of where the mosquitoes fed (i.e. blood meal source). That was not shown so as to give the=at strong evidence.

This study can compare the enzyme changes and resistance and enzyme presentation between lowland and highland but not between indoor and outdoor resting mosquitoes

More specific comments are uploaded in attached document.

6. PLOS authors have the option to publish the peer review history of their article (what does this mean?). If published, this will include your full peer review and any attached files.

Reviewer #1: No

Reviewer #2: No

---

## [Author Response · Author response to Decision Letter 0]

13 Jan 2021

Response to reviewer comments

The following are our point-by-point responses to the comments raised by the reviewers. We really thank the reviewers for their constructive criticism and insights that have helped to improve this paper.

Reviewer Comments:

Reviewer #1

Reviewer: Line 73 to 75 « Despite the observed improvement in malaria incidence and prevalence in many parts of sub-Saharan Africa, transmission is increasing in several countries [2,3]. » I don’t think the term « improvement » is appropriate and the sentence doesn’t make sense 

Response: The sentence has been revised and now reads” Despite the observed achievements in malaria reduction, many parts of sub-Saharan Africa still suffer greatly from the disease”….Line 72-74

Reviewer: Line 80 to 81 « Insecticide resistance in malaria mosquitoes is linked to presence and increase in metabolic detoxification enzymes, target site insensitivity and behavioural resistance [11]. » the sentence is not clear can the author rephrase the sentence to improve understanding?

Response: The sentence has been revised to read “Insecticide resistance in malaria mosquitoes has been linked to target-site insensitivity, elevated levels of metabolic detoxifying enzymes”……Line 77-78

Reviewer: Line 84 to 86 « Detoxification enzyme systems that have been reported to confer resistance include three major families of enzymes; the cytochrome P450monooxygenases, β-esterases, and the Glutathione S-transferases. » please write esterase as you are refering to enzymes families

Response: This has been addressed …Line 82. 

Reviewer: Line 92 to 95 « The increasing levels of insecticide resistance in malaria mosquitoes have been associated with continuous exposure to insecticides in LongLasting Insecticide Nets( LLINs) [23,24] and agro-chemicals such as pesticides due to the creation of selection pressures [25-27]. » The phrase is not clear and need to be rewritten to improve understanding

Response: The statement has been rewritten and now it reads “ The increasing levels of insecticide resistance in malaria mosquitoes is believed to be mainly caused by scaling up of insecticidal treated nets (ITNs) [23,24] and indiscriminate use of agro-chemicals for controlling crop pests in agriculture[25-27] ..Line 88-90

Reviewer: The introduction is not straight forward and is poorly written the authors need to revise this section carefully to improve understanding

Response: The introduction section has been revised to improve understanding.

Reviewer: Line 120 « The study was carried out in the lowland site of Kisian (0.0749° S, 34.6663° E, 1,137m) in Kisumu county … » The geographical coordinates is not appropriate can the authors check for the correct coordinates ?

Response: The geographical coordinates provided for Kisian are the correct one “00.0749° S, 034.6663° E, altitude 1,137-1,330 m above sea level)”

Reviewer: Line 127 to 128 « There is extensive use of agrochemicals on these farms which could have a potential role in the mediation of resistance to insecticides. » Are anopheline breeding habitats found in tobacco farms and what type of insecticides are used in these farms please provide more information. Please also say how frequent insecticides are sprayed in these farms. How distant are these farms from the village.

Response: The sentence has been modified from line 120 -125 and also cited Wanjala et al 2015 which has detailed information on the agrochemicals used in the study area.

Reviewer: Mosquitoes were collected in how many houses per site and outdoor places please include information.

Response: The information on the number of houses sampled and outdoor places has been included under mosquito sampling section …. ‘Thirty (30) houses were randomly selected per site and resting mosquitoes collected from 06:00 to 09:00 h……” Line 130-139

Reviewer: Line 156 to 160 « First filial generation (F1) females raised from field-collected adults that were resting either indoors or outdoors, that were 3-5 -day old, were tested for susceptibility using the standard WHO tube bioassays (WHO, 2016) against discriminating doses of five insecticides selected from three classes: (i)Pyrethroids - (0.05% deltamethrin, 0.75% permethrin and (0.05% Alphacypermethrin); and (ii) organophosphate - (5% malathion). » The author say they tested 5 insecticides only a list of four insecticide is provided please correct.

Response: This has been corrected in the text, four insecticides from 2 classes of insecticides were used Line161.

Reviewer: Line 166 to 167 « Mortality was defined as the inability of the mosquitoes to stand or to fly in a coordinated manner. » is this mortality or knock down effect please correct.

Response: The statement has been deleted from the text.

Reviewer: Line 175 to 176 « After pre-exposure to PBO, the mosquitoes were immediately exposed to the three pyrethroids (deltamethrin, permethrin and alphacypermethrin) for an additional hour. » Where mosquitoes exposed to all three insecticides at once or seperately to each insecticide please clarify

Response: The sentence has been corrected and now reads “After pre-exposure to PBO, the mosquitoes were immediately exposed to each of the three pyrethroids (deltamethrin, permethrin and alphacypermethrin) separately for another hour….Line 177-178

Reviewer: There are data missen in this part the author need to give the number of gravid and non gravid females collected outdoor and indoor in each site. They could alsoindicate the number collected during each season.

Response: We have included the data for the gravid females that were used for the study from line 155-157. 

Reviewer: The number of mosquitoes tested for each insecticide need to be provided

Responses: The number of mosquitoes tested for each insecticide was 150 according to the WHO 2016 Insecticide resistance testing procedures the information is already provide in the text.. Line 163

Reviewer: The authors say that the percentage is in bracket but it is not clear how they did the calculations of the percentage of An. funestus in Kisian 122(19%) in Kimaeti they also wrote 167 (23.16) can they provide explanation for their calculations or correct. The author also need to check their calculations the row on An. arabiensis in Kisian the total in the column and line also need to be corrected.

Response: The figures in table 1 has been corrected

Reviewer #2

This study intends to show the insecticides resistance between indoors and outdoors resting vectors in highlands and lowlands This study has a major drawback in methodology. To start with the ecological behavior of An. gambiae s.l is to feed indoor and rest indoor for An. gambiae s.s. while for An. arabienses is feeding indoor and resting outdoor or feed outdoor and rest outdoors. Due to the wide coverage of the insecticide treated surfaces (LLINs and RS), mosquitoes have changed the feeding and resting behavior to forfeit the effect of the insecticides treated surfaces. Due to this, higher proportions have found to feed and rest outdoors regardless of the vector species.

Sampling: The samplings of vectors were done indoor and outdoor. The vector after feeding indoor they get out to rest and search for oviposition site. This is indeed achallenge to mark that this mosquito was found resting outdoor while it fed indoor. The physiological and enzymatic dynamics in mosquitoes is mostly reflected byblood meal source. There is no barrier between indoor and outdoor resting populations hence free mating is possible. In first place, authors should be aware thatresistance is genetical and is clearly related age of the mosquito. These mosquitoes share breeding sited and feeding sources. The emerging young mosquitoesresting outdoor before searching for host. The outdoor collection is probably that it collected emerged and gravid mosquitoes searching for oviposition site. Themethods were weak not to show the separation of population fed outdoor or indoor. So collection alone cannot provide a guarantee of the biochemical processes tobe elevated by resting position. I personally reject this paper as the basis used of outdoor and indoor resting have no barrier separation between the populations collected. Since resistance isgenetical and biochemical process is influenced by blood meal sources. No evidence of where the mosquitoes fed (i.e. blood meal source). That was not shown so as to give the=at strong evidence.

This study can compare the enzyme changes and resistance and enzyme presentation between lowland and highland but not between indoor and outdoor resting mosquitoes More specific comments are uploaded in attached document.

Response: Our study looked at the differences in insecticide resistance between indoor and outdoor resting mosquitoes. The focus was not on biting mosquitoes as mosquitoes are likely to be influenced by where the host might be, whether indoor or outdoor, to take a blood meal. They are likely to make a choice, after taking a blood meal, between resting indoors or outdoors based on their preferences as exophilic or endophilic vectors. We acknowledge that the taking of a blood meal by mosquitoes might influence their insecticide resistance status through the raising of biochemical enzymes to tolerate insecticides. This made us not to use the female mosquitoes collected but their F1 progeny so that we can understand whether resting behaviour is a consistent behaviour. We have clarified our objective and some parts of the methodology, which should make our methodology clear to answer the points raised by the reviewer 2. We strongly disagree with the reviewer that we should have shown where the mosquitoes took their blood meals. We also reject the suggestion that we “can compare the enzyme changes and resistance and enzyme presentation between lowland and highland”. This has no scientific basis. The methodology in our study was not weak as alleged by the reviewer but probably he may not have clearly understood it. 

Editorial comments

To the editor:

Figure 1 image has been removed from the manuscript. 

We have also taken the ethics statement to the methods section

---

## [Decision Letter · Decision Letter 1]

16 Feb 2021

Insecticide resistance status of indoor and outdoor resting malaria vectors in a highland and lowland site in Western Kenya

PONE-D-20-30800R1

Dear Dr. Afrane,

We’re pleased to inform you that your manuscript has been judged scientifically suitable for publication and will be formally accepted for publication once it meets all outstanding technical requirements.

Kind regards,

Basil Brooke, PhD

Academic Editor

PLOS ONE

Additional Editor Comments (optional):

Reviewers' comments:

Reviewer's Responses to Questions

**Comments to the Author**

1. If the authors have adequately addressed your comments raised in a previous round of review and you feel that this manuscript is now acceptable for publication, you may indicate that here to bypass the “Comments to the Author” section, enter your conflict of interest statement in the “Confidential to Editor” section, and submit your "Accept" recommendation.

Reviewer #2: All comments have been addressed

2. Is the manuscript technically sound, and do the data support the conclusions?

Reviewer #2: Yes

3. Has the statistical analysis been performed appropriately and rigorously? 

Reviewer #2: Yes

4. Have the authors made all data underlying the findings in their manuscript fully available?

Reviewer #2: Yes

5. Is the manuscript presented in an intelligible fashion and written in standard English?

Reviewer #2: Yes

6. Review Comments to the Author

Reviewer #2: I have realized that, all comments I raised in first place were all addressed in this revised version.

7. PLOS authors have the option to publish the peer review history of their article (what does this mean?). If published, this will include your full peer review and any attached files.

Reviewer #2: No

---

## [Editor Report · Acceptance letter]

19 Feb 2021

PONE-D-20-30800R1 

Insecticide resistance status of indoor and outdoor resting malaria vectors in a highland and lowland site in Western Kenya 

Dear Dr. Afrane:

I'm pleased to inform you that your manuscript has been deemed suitable for publication in PLOS ONE. Congratulations! Your manuscript is now with our production department. 

Kind regards, 

on behalf of

Dr Basil Brooke 

Academic Editor

PLOS ONE